# Electrocardiographic Predictors of Primary Ventricular Fibrillation and 30-Day Mortality in Patients Presenting with ST-Segment Elevation Myocardial Infarction

**DOI:** 10.3390/jcm10245933

**Published:** 2021-12-17

**Authors:** Alberto Cipriani, Gianpiero D’Amico, Giulia Brunetti, Giovanni Maria Vescovo, Filippo Donato, Marco Gambato, Pietro Bernardo Dall’Aglio, Francesco Cardaioli, Martina Previato, Nicolò Martini, Martina Perazzolo Marra, Sabino Iliceto, Luisa Cacciavillani, Domenico Corrado, Alessandro Zorzi

**Affiliations:** 1Department of Cardiac, Thoracic and Vascular Sciences, University of Padua, 35128 Padova, Italy; alberto.cipriani@unipd.it (A.C.); giulia.brunetti@studenti.unipd.it (G.B.); filippodonato89@gmail.com (F.D.); marco.gambato@unipd.it (M.G.); pietrobdallaglio@gmail.com (P.B.D.); francesco.cardaioli@gmail.com (F.C.); nicolo.martini.2@studenti.unipd.it (N.M.); martina.perazzolomarra@unipd.it (M.P.M.); sabino.iliceto@unipd.it (S.I.); luisa.cacciavillani@unipd.it (L.C.); alessandro.zorzi@unipd.it (A.Z.); 2Department of Cardiology, Ospedale dell’Angelo, 30174 Venice, Italy; gianpiero.damico@hotmail.it (G.D.); gm.vescovo@libero.it (G.M.V.); martina.prev@gmail.com (M.P.)

**Keywords:** acute myocardial infarction, electrocardiogram, ST-segment elevation, ventricular fibrillation

## Abstract

Primary ventricular fibrillation (PVF) may occur in the early phase of ST-elevation myocardial infarction (STEMI) prior to primary percutaneous coronary intervention (PCI). Multiple electrocardiographic STEMI patterns are associated with PVF and short-term mortality including the tombstone, Lambda, and triangular QRS-ST-T waveform (TW). We aimed to compare the predictive value of different electrocardiographic STEMI patterns for PVF and 30-day mortality. We included a consecutive cohort of 407 STEMI patients (75% males, median age 66 years) presenting within 12 h of symptoms onset. At first medical contact, 14 (3%) showed the TW or Lambda ECG patterns, which were combined in a single group (TW-Lambda pattern) characterized by giant R-wave and downsloping ST-segment. PVF prior to primary PCI occurred in 39 (10%) patients, significantly more often in patients with the TW-Lambda pattern than those without (50% vs. 8%, *p* < 0.001). For the multivariable analysis, Killip class ≥3 (OR 6.19, 95% CI 2.37–16.1, *p* < 0.001) and TW-Lambda pattern (OR 9.64, 95% CI 2.99–31.0, *p* < 0.001) remained as independent predictors of PVF. Thirty-day mortality was also higher in patients with the TW-Lambda pattern than in those without (43% vs. 6%, *p* < 0.001). However, only LVEF (OR 0.86, 95% CI 0.82–0.90, *p* < 0.001) and PVF (OR 4.61, 95% CI 1.49–14.3, *p* = 0.042) remained independent predictors of mortality. A mediation analysis showed that the effect of TW-Lambda pattern on mortality was mediated mainly via the reduced LVEF. In conclusion, among patients presenting with STEMI, the electrocardiographic TW-Lambda pattern was associated with both PVF before PCI and 30-day mortality. Therefore, this ECG pattern may be useful for early risk stratification of STEMI.

## 1. Introduction

Primary ventricular fibrillation (PVF) occurs commonly during the early phase of ST-elevation myocardial infarction (STEMI) before primary percutaneous coronary intervention (PCI), and has been associated with an increased short- [1] and long-term risk of death [2,3].

Several risk factors for PVF are known including clinical, hemodynamic, and electrocardiographic parameters, and their recognition is useful to optimize the early management of STEMI patients [4,5,6]. Besides its diagnostic value and the information regarding the culprit coronary vessel and extension of the ischemic area, the electrocardiogram (ECG) may identify subgroups of patients at higher risk of acute complications. In particular, multiple ECG patterns have been found to be associated with PVF [7,8,9,10,11]. Recently, we described the “triangular QRS-ST-T waveform” (TW), a distinctive STEMI pattern characterized by a single, giant, triangular wave, resulting from the fusion of the QRS complex, the ST-segment, and the T-wave, and we observed that patients presenting with the TW pattern more frequently had PVF, cardiogenic shock, and higher in-hospital mortality [10]. Similarly, the “lambda-like” pattern, characterized by an ST elevation resembling the Greek letter lambda, was described in patients with acute myocardial infarction (AMI) complicated with multiple episodes of polymorphic ventricular tachycardia and PVF [8]. These STEMI patterns have two features in common: the giant R wave (amplitude > 1 mV) and the steep downsloping of the ST segment, each of which have been described in association with PVF [9,11]. However, a specific quantitative risk analysis has never been performed.

The aim of the study was to assess the predictive value of different electrocardiographic patterns for the occurrence of PVF during the early phase of STEMI and for 30-day mortality.

## 2. Materials and Methods

In this observational single-center study, we enrolled all consecutive patients admitted to the Cardiology Unit of the University Hospital of Padua with a diagnosis of STEMI from January 2015 to July 2017. Data were collected from in-hospital and outpatient clinical evaluations and digital medical records of our hospital. The investigation was carried out following the rules of the Declaration of Helsinki of 1975; the ethics review board of our institution was notified of the study protocol. Given the retrospective and observational nature of the study, consent from the patients was not required.

Eligible patients were all STEMI patients with a readable 12-lead ECG at the time of the first medical contact (FMC). According to the 2012 European Society of Cardiology Guidelines [12], STEMI was diagnosed in the case of (1) symptoms consistent with myocardial ischemia; and (2) ST-segment elevation measured at the J point in at least two contiguous leads, ≥0.25 mV in men below the age of 40 years, ≥0.2 mV in men over the age of 40 years, or ≥0.15 mV in women in leads V2–V3 and/or ≥0.1 mV in other leads, in the absence of left ventricular hypertrophy or left bundle branch block, or ST-segment depression in lead V1–V3 confirmed by concomitant ST-segment elevation ≥0.05 mV recorded in leads V7–V9.

Data systematically collected included anamnestic data (especially those of special interest for cardiological risk stratification), Killip class on first medical contact, and ECG. The 12-lead ECGs (25 mm/s, 10 mm/mV, 0.05–150 Hz) were acquired in most cases in the pre-hospital setting by the emergency medical service for early identification of STEMI and were prospectively collected and analyzed by two investigators (M.G., G.M.V.); controversial cases were solved by consensus. The analysis focused on particular ECG patterns including the “tombstone”, “lambda”, and “TW”, which were defined according to previous publications [8,10,13]. Figure 1 provides a schematic representation of these STEMI ECG patterns. The TW and Lambda patterns were merged in a single group (TW-Lambda pattern) as proposed by Aizawa et al. [9]. Giant R wave was defined as R amplitude ≥1.0 mV.

All patients underwent an echocardiography at admission to evaluate left ventricular ejection fraction (LVEF). The GRACE score on admission was calculated for all patients [14]. Two endpoints were considered: PVF (VF occurring before invasive coronary angiography and PCI) and mortality at 30 days.

Statistical analysis was performed using R-software Version 3.3.3 for Mac OS X (R Foundation for Statistical Computing, Vienna, Austria). As normality could not be assumed for any variables, continuous variables were presented as median (1st–3rd quartiles) and compared with non-parametric tests such as the Mann–Whitney test. Categorical variables were reported as number and percentage and compared using the χ^2^ test or Fisher’s exact test, as appropriate. Univariate logistic regression analysis was performed to investigate the relationship between the outcomes (PVF and 30-day mortality) and (1) variables that showed significant differences in medians or proportions between groups; and (2) other relevant variables based on previous study [4]. Among the significant variables at univariate analysis (*p* < 0.05), for each outcome, we chose to enter into the multivariate logistic regression model the ones that we considered more clinically relevant for the scope of the study. The number of variables that were chosen was limited in order to maintain a ≈1:10 co-variates to outcome ratio to avoid the risk of overfitting. We then used a stepwise selection approach based on Akaike information criterion (AIC) to improve the model’s performance. A mediation analysis (Causal Mediation Analysis, package ‘mediation’ version 4.5.0, R-software Version 3.3.3, Vienna, Austria) was performed, to quantify the extent to which the PVF and LVEF ejection fraction participated in the relationship between the TW-Lambda pattern and 30-day mortality. We searched for the significance of the indirect effect of the independent variable on the dependent one using bootstrapping procedures. Unstandardized indirect effects were computed for each of 1000 bootstrapped samples, and the 95% confidence interval was computed by determining the indirect effects at the 2.5th and 97.5th percentiles. A two-sided *p* value of <0.05 was considered indicative of statistical significance.

## 3. Results

The study population included 407 patients (307 males, 75%) with a median age of 66 (56–75) years. The ECG of the first medical contact was recorded after a median of 169 (41–200) minutes from symptom onset, showing ST-segment elevation in anterior leads V1–V4 in 189 (46%) and in inferior leads II/aVF/III in 117 (29%). Left main coronary artery was the culprit vessel in 21 (5%) patients and left ventricular descending artery in 197 (48%). Primary PCI was performed in 346 (85%). Considering ECG STEMI patterns, 45 (11%) patients showed the tombstone pattern and 14 (3%) the TW-Lambda pattern. Giant R wave was observed in 125 (31%). Baseline characteristics, risk factors, clinical, and electrocardiographic features are depicted in Table 1.

Primary ventricular fibrillation prior to PCI occurred in 39 (10%) patients. Table 2 shows the distribution of anamnestic, clinical, and ECG features among patients with or without PVF. PVF prior to primary PCI occurred in 39 (10%) patients, significantly more often in patients with the TW-Lambda pattern than those without (50% vs. 8%, *p* < 0.001). Of note, patients with the TW-Lambda pattern also showed a lower LVEF at admission (median 32%) than those without (48%, *p* < 0.001).

Univariable logistic regression analysis showed that age, Killip class ≥3, anterior STEMI, giant R wave, and TW-Lambda pattern emerged as significant risk factors for PVF (Table 3).

Stepwise multivariate logistic regression analysis included four relevant variables, in other words, three ECG features (TW-Lambda pattern, giant R waves, anterior STEMI) and hemodynamic status (Killip class), Killip class ≥3 (OR 6.19, 95% CI 2.37, 16.1, *p* < 0.001), and TW-Lambda pattern (OR 9.64, 95% CI 2.99, 31.0, *p* < 0.001) remained independent predictors of PVF (Table 3).

At 30-day follow up, we recorded 28 deaths (7%). Table 4 shows the distribution of different variables in patients who died at 30 days. Mortality was significantly higher in patients with the TW-Lambda pattern than in those without (43% vs. 6%, *p* < 0.001). An increased risk of death was also identified for PVF, EF during hospitalization, GFR, arterial hypertension, Grace score, Killip class ≥3.

Univariable logistic regression analysis showed that age, arterial hypertension, Grace score, Killip class ≥3, GFR, LVEF, peak Troponin I, PVF, and TW-Lambda pattern emerged as significant risk factors for 30-day death (Table 5).

After performing multivariate stepwise logistic regression analysis including the three most clinically (and rapidly assessable at patient’s admission) relevant factors (such as LVEF, Killip class, PVF, and TW-Lambda pattern), only LVEF (OR 0.86, 95% CI 0.82, 0.90, *p* < 0.001) and PVF (OR 4.61, 95% CI 1.49, 14.3, *p* = 0.042) were maintained in the regression model (Table 5).

Thus, a mediation analysis was performed to analyze the possible mediation effect of these latter variables in determining univariate significance of the ECG pattern on 30-day mortality (Table 6 and Table 7). The analysis confirmed that the effect of TW-Lambda pattern on 30-day mortality was mediated via the presence of reduced LVEF, but not of PVF. Indeed, the bootstrapped unstandardized indirect effect (i.e., the effect of TW-Lambda on 30-day mortality going through the considered mediators) was tested to be significant only when considering LVEF as a mediator (estimate 0.235 95% CI 0.113, 0.350, <0.001), though a trend toward significance was also found when considering PVF as the mediator (estimate 0.063, 95% CI −0.013, 0.140, *p* = 0.108).

## 4. Discussion

In the present study, we investigated the predictive value of specific electrocardiographic STEMI patterns—the tombstone, Lambda, and TW (Figure 1)—for the occurrence of PVF during the early phase of STEMI and for 30-day mortality. The main results of the study were: (1) ECG detection of a Lambda or TW STEMI pattern was a strong and independent predictor of PVF before primary PCI; (2) significant predictors of 30-day mortality were the occurrence of PVF, and lower LVEF; and (3) the effect of Lambda or TW STEMI pattern on 30-mortality was significantly mediated by a lower LVEF.

The ECG has a central role in the early management of AMI as it helps in providing diagnosis and guiding the appropriate therapy [15]. Additionally, the ECG can offer the potential to provide helpful information about the prognosis of AMI patients, in terms of arrhythmic or hemodynamic complications and mortality. The tombstone, Lambda, and TW STEMI patterns represent severe QRS-ST-T deformations that have been previously associated with PVF and poor outcome [9,10,12]. However, most of these observations are anecdotal and their predictive value for PVF and mortality has never been analyzed. Aizawa et al. classified the ECG of STEMI patients into three main types and found that the “Type 1” (namely the TW-Lambda pattern), defined as a QRS-ST-T pattern characterized by a downsloping J-ST segment toward T waves immediately after the R wave, without a flat or rising portion, was highly associated with PVF [11] (Figure 2). In keeping with this, our study demonstrated that the TW-Lambda pattern is an independent predictor of PVF, conferring to the STEMI patients a 6-fold increase in risk of PVF before primary PCI. The steep downsloping of the J-ST-segment appears to be the common denominator of these high-risk patterns and may be the expression of the diffuse dispersion of excitability, conduction, and refractoriness occurring in the myocardium during a severe and extensive ischemia, predisposing to PVF triggered by R-on-T phenomena (phase-2 re-entry) such as those observed in Brugada syndrome [16,17].

Other significant predictors of PVF were the younger age, a Killip class ≥3, the giant R-wave, and the anterior AMI. A trend toward an inverse relationship between age and PVF was also observed in a recent paper [18] and in a previous metanalysis [19], and may be related to the fact that younger STEMI patients more often suffer an acute coronary artery occlusion without pre-conditioning or collateral flow. The anterior location of AMI has also been found to be a major risk factor of PVF in previous investigations [20,21]. This finding may confirm that the ischemia of the anterior wall has a greater propensity for PVF, being more densely innervated by cardiac sympathetic fibers, which in the presence of ischemia may lead to a greater release of catecholamines and trigger ventricular arrhythmias [22,23].

In the pre-PCI era, the tombstone pattern, as its name suggests, was a marker of severe ischemia and extensive cardiac damage and was associated with fatal arrhythmias and poor outcome [7,13]. However, our data showed that the tombstone pattern was neither associated with PVF nor with higher mortality. This may be due to the devolvement of the primary PCI network, which enables aa reduction in reperfusion time, thereby reducing the occurrence of electrical and mechanical complications and improving outcome [24].

In our study, PVF before primary PCI occurred in 10% of STEMI patients, a prevalence consistent with recent reports [20,25]. Likewise, the 30-day mortality for STEMI in our hospital (28/407, 7%) was comparable to that of another region in Italy [26] and within the range of those reported for the U.S. and European populations [27,28,29,30]. The second endpoint of the study, 30-day mortality, was associated with the TW-Lambda pattern as well as PVF and lower LVEF, a finding in keeping with previous reports [31]. However, only PVF and lower LVEF, but not the TW-Lambda ECG pattern, remained independent predictors of death in the multivariable analysis. This is not surprising considering that the TW-Lambda is not a physiopathological mechanism directly linked to increased mortality, but rather an epiphenomenon of increased electrical instability (higher risk of PVF, as discussed above) and higher amount of ischemic myocardium (hence worse LVEF at admission). The fact that the occurrence of the TW-Lambda pattern often reflects an acute occlusion of a proximal coronary artery justifies why these patients typically show a severe impairment of LV systolic function [10]. Interestingly, the indirect effect of the TW-Lambda ECG pattern on mortality was predominantly mediated by reduced LVEF.

According to these findings, the observation of a TW-Lambda pattern should influence the management of STEMI patients, not only suggesting the need for a stricter pre-PCI monitoring, but also of a more aggressive management, for example, referring the patient to a hospital able to deliver a high-intensity level of care (e.g., capability of ventricular assist device implantation).

The limitations of the study include its observational nature and the fact that patients who died outside of hospitals were not included in the analysis; thus, our results may not be generalizable to all STEMI patients.

In conclusion, this study showed that STEMI patients presenting with the TW-Lambda pattern have a higher risk of PVF and 30-day mortality. These results have clinical implications in improving the early arrhythmic risk stratification of STEMI patients and in planning an adequate therapeutic strategy.

## Figures and Tables

**Figure 1 jcm-10-05933-f001:**
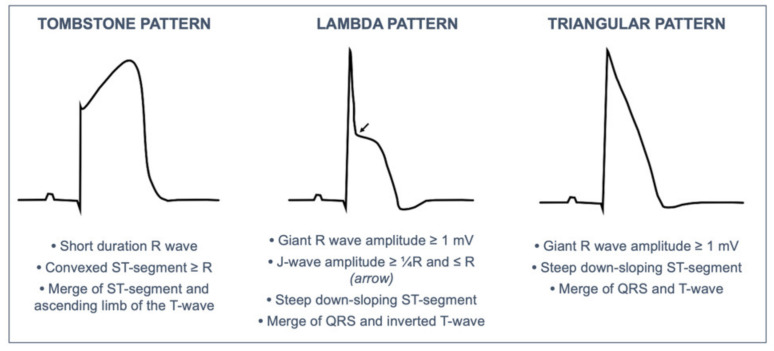
Definition of the electrocardiographic STEMI pattern analyzed in the study.

**Figure 2 jcm-10-05933-f002:**
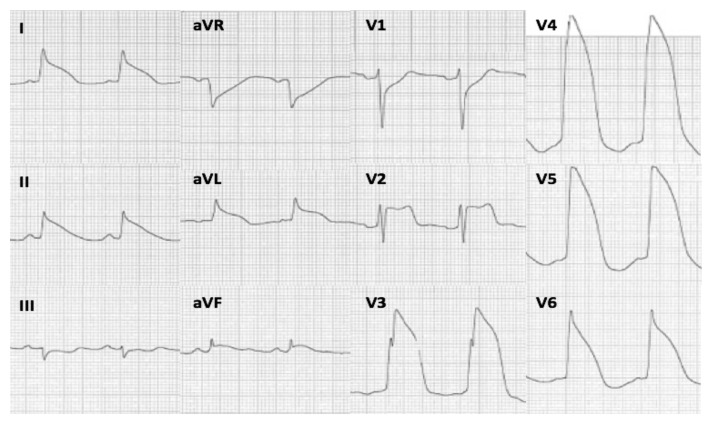
Electrocardiogram of a 68-year-old male patient who presented to the emergency department for chest pain. A triangular QRS-ST-T waveform was appreciable on anterolateral and inferior leads. Shortly after the ECG recording, the patient had PVF, requiring urgent defibrillation. Coronary angiography showed an occlusion of the proximal left anterior descending coronary artery.

**Table 1 jcm-10-05933-t001:** Clinical and electrocardiographic features in the study population. Values are expressed as number of patients (%) or median [25 and 75% percentiles].

	Study Population (*n* = 407)
Male sex	307 (75)
Age, years	66 (56–75)
FMC to PCI, min	101 (55–120)
Primary PCI	346 (85)
**Medical history**	
Arterial hypertension	261 (64)
Dyslipidemia	176 (43)
Diabetes mellitus	75 (18)
Familiar CAD	153 (38)
Smoke	139 (34)
Previous AMI	35 (9)
CKD	36 (9)
Severe COPD	13 (3)
**ECG at first medical contact**	
Anterior STEMI (V1–V4)	189 (46)
Anterior (V1–V4) and lateral (V5- V5–V6, I, aVL V6, I, aVL)	23 (6)
Lateral (V5–V6, I, aVL)	14 (3)
Inferior (II/aVF/III)	117 (29)
Inferior (II/aVF/III) and lateral (V5–V6, I, aVL)	59 (14)
Posterior	5 (1)
Tombstone pattern	45 (11)
Triangular-Lambda pattern	14 (3)
R-wave ≥ 1.0 mV	125 (31)
**Culprit coronary vessel**	
Left main	21 (5)
LAD	197 (48)
Left circumflex	52 (13)
Ramus intermedius	3 (1)
Right coronary	134 (33)
**Clinical variables**	
LVEF, %	47 (40–55)
GFR, mL/min/1.73 m^2^	75 (61–90)
Killip class ≥ III	24 (6)
Peak Troponin I, ug/L	70 (26–150)
**Events**	
PVF	39 (10)
30-day mortality	28 (7)

AMI = acute myocardial infarction; CAD = coronary artery disease; CKD = chronic kidney disease; COPD = chronic obstructive pulmonary disease; FMC = first medical contact; GFR = glomerular filtration rate; LAD = left anterior descending; LVEF = left ventricular ejection fraction; PCI = percutaneous coronary intervention; PVF = primary ventricular fibrillation.

**Table 2 jcm-10-05933-t002:** Distribution of anamnestic, clinical, and electrocardiographic features among the groups with and without PVF.

	No PVF *n* = 368 (90%)	PVF *n* = 39 (10%)	*p* Value
**Anamnestic and clinical variables**			
Age, years	65.4 (52.1, 78.5)	61.2 (50.2, 73.3)	0.033 *
Sex, male	273 (74.2)	34 (87.2)	0.073
Arterial hypertension	236 (64.1)	25 (64.1)	0.563
Dyslipidemia	157 (42.7)	17 (43.6)	0.288
Diabetes mellitus	68 (18.5)	7 (17.9)	0.569
Familiar CAD	141 (38.3)	12 (30.8)	0.228
Smoking	122 (33.2)	17 (43.6)	0.130
Severe COPD	12 (3.3)	1 (2.6)	0.641
CKD	33 (9.0)	3 (7.7)	0.539
Previous AMI	30 (8.2)	5 (12.8)	0.234
Killip class ≥3	15 (4.1)	9 (23.1)	<0.001 *
**ECG variables**			
New onset AF	24 (6.5)	5 (12.8)	0.132
Advanced AV block	11 (3.0)	1 (2.6)	0.678
Anterior STEMI	163 (44.3)	25 (64.1)	0.014 *
Lateral STEMI	21 (5.7)	2 (5.1)	0.238
Inferior STEMI	108 (29.3)	9 (23.1)	0.267
Posterior STEMI	5 (1.4)	0 (0)	-
Tombstone pattern	38 (10.3)	7 (17.9)	0.175
Triangular-Lambda pattern	7 (1.9)	7 (17.9)	<0.001 *
Giant R wave	107 (29.1)	18 (46.2)	0.024 *

* means *p* < 0.050. Values are expressed as number of patients (%) or median [25 and 75% percentiles]. AF = atrial fibrillation; AMI = acute myocardial infarction; AV = atrio-ventricular block; CAD = coronary artery disease; CKD = chronic kidney disease; COPD = chronic obstructive pulmonary disease; ECG = electrocardiography; FMC = first medical contact; LAD = left anterior descending; LVEF = left ventricular ejection fraction; PCI = percutaneous coronary intervention; STEMI = ST-elevated myocardial infarction; PVF = primary ventricular fibrillation.

**Table 3 jcm-10-05933-t003:** Predictors of primary ventricular fibrillation at logistic regression analysis.

Primary Ventricular Fibrillation	Univariate Logistic Regression			Multivariate Logistic Regression		
	OR	95% CI	*p*	OR	95% CI	*p*
Age	0.74	0.57, 0.97	0.030			
Arterial hypertension	0.99	0.50, 1.99	0.997			
Diabetes Mellitus	0.76	0.41, 2.28	0.935			
Killip class ≥ 3	7.06	2.85, 17.50	<0.001	6.19	2.37, 16.1	0.035
Anterior STEMI	2.26	1.15, 4.59	0.021			
Triangular-Lambda wave	11.3	3.72, 34.2	<0.001	9.64	2.99, 31.0	0.027
R-wave ≥ 1.0 mV	2.09	1.06, 4.08	0.030			

STEMI = ST-segment elevation myocardial infarction.

**Table 4 jcm-10-05933-t004:** Distribution of anamnestic, clinical, and instrumental variables among the groups with and without death at 30 days. * means *p* < 0.050. Values are expressed as number of patients (%) or median [25 and 75% percentiles].

	Alive *n* = 379 (93%)	Dead *n* = 28 (7%)	*p* Value
**Anamnestic variables**			
Age, years	64.6 (51.4, 79.1)	70.1 (56.7, 83.2)	0.043 *
Male sex	284 (74.9)	23 (82.1)	0.273
Arterial hypertension	238 (62.8)	23 (82.1)	0.028 *
Dyslipidemia	168 (44.3)	8 (28.6)	0.075
Diabetes mellitus	66 (17.4)	9 (32.1)	0.073
Familiar CAD	148 (39.1)	5 (17.9)	0.180
Smoking	129 (34.0)	10 (35.7)	0.503
Previous AMI	34 (9.0)	1 (3.6)	0.283
CKD	32 (8.4)	4 (14.3)	0.227
Severe COPD	10 (2.6)	3 (10.7)	0.052
**Clinical and instrumental variables**			
Killip class ≥3	15 (4.0)	9 (32.1)	<0.001 *
Grace score	140 (18, 210)	191 (60, 250)	<0.001 *
GFR mL/min/1.73 m^2^	81 (62, 90)	45 (36, 75)	<0.001 *
Peak Troponin I, ug/L	67.2 (1, 579)	115 (22, 400)	0.005 *
Delayed FMC-PCI > 120 min	39 (10.3)	5 (17.9)	0.283
Culprit lesion			
Left main	16 (4.7)	4 (22.2)	0.013 *
LAD	181 (52.6)	11 (61.1)	0.324
Left Circumflex	45 (13.1)	5 (27.8)	0.086
RCA	127 (36.9)	3 (16.7)	0.063
Ramus Intermedius	2 (0.6)	1 (5.6)	0.142
EF (%)	49 (20, 71)	26.5 (9, 54)	<0.001 *
Tombstone pattern	40 (10.6)	5 (17.9)	0.218
Triangular-Lambda pattern	8 (2.1)	6 (21.4)	<0.001 *
Giant R wave	114 (30.1)	11 (39.3)	0.208
PVF	31 (8.2)	8 (28.6)	0.003 *
AF	24 (6.3)	5 (17.9)	0.054
Advanced AV block	10 (2.6)	2 (7.1)	0.197

AF = atrial fibrillation; AMI = acute myocardial infarction; AV = atrio-ventricular block; CAD = coronary artery disease; CKD = chronic kidney disease; COPD = chronic obstructive pulmonary disease; ECG = electrocardiography; FMC = first medical contact; GFR = glomerular filtration rate; LAD = left anterior descending; LVEF = left ventricular ejection fraction; PCI = percutaneous coronary intervention; STEMI = ST-elevated myocardial infarction; PVF = primary ventricular fibrillation.

**Table 5 jcm-10-05933-t005:** Predictors of 30-day mortality at logistic regression analysis.

30 Days Death	Univariate Logistic Regression			Multivariate Logistic Regression		
	OR	95% CI	*p*	OR	95% CI	*p*
Age	1.03	1.00, 1.07	0.036 *			
Arterial hypertension	2.73	1.01, 7.33	0.047 *			
Grace score	1.03	1.02, 1.04	<0.001 *			
Killip class ≥3	11.5	4.46, 29.6	<0.001 *			
GFR	0.96	0.94, 0.97	<0.001 *			
LVEF	0.86	0.82, 0.90	<0.001 *	0.86	0.82, 0.90	<0.001 *
Peak Troponin I	1.02	1.01, 1.08	0.012 *			
PVF	4.49	1.63, 11.0	0.001 *	4.61	1.49, 14.3	0.042 *
Triangular/Lambda pattern	12.6	4.03, 39.6	<0.001 *			

* means *p* < 0.050. GFR = glomerular filtration rate (mL/min/1.73 m^2^); LVEF = left ventricular ejection fraction; PVF = primary ventricular fibrillation.

**Table 6 jcm-10-05933-t006:** Mediation analysis exploring the effect of triangular/lambda pattern on 30-day mortality mediated by primary ventricular fibrillation.

Effect	Estimate	95% CI	*p*	% Mediation
Indirect	0.063	−0.013, 0.140	0.108	17.0
Direct	0.305	0.058, 0.590	0.016 *	83.0
Total	0.369	0.088, 0.650	0.008 *	100.0

* means *p* < 0.050.

**Table 7 jcm-10-05933-t007:** Mediation analysis exploring the effect of triangular/lambda pattern on 30-day mortality mediated by left ventricular function.

Effect	Estimate	95% CI	*p*	% Mediation
Indirect	0.235	0.113, 0.350	<0.001	62.7
Direct	0.139	−0.039, 0.360	0.112	37.3
Total	0.374	0.114, 0.640	0.008 *	100.0

* means *p* < 0.050.

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
