# Peer review of "Electrocardiographic Predictors of Primary Ventricular Fibrillation and 30-Day Mortality in Patients Presenting with ST-Segment Elevation Myocardial Infarction"

_jcm, 2021, doi:10.3390/jcm10245933_

Round 1

Reviewer 1 Report

The manuscript is generally interesting. The study design is good and several diagnostic methods were used. I have following comments:

1) In my opinion, additional laboratory markers should be considered as predictors.

2) Is it justified to consider glomerular filtration rate as a predictor of ventricular fibrillation?

3) P-values should be consistently reported to 3 decimals.

4) More than 70% of references are over 10 years old.

Author Response

  1. The manuscript is generally interesting. The study design is good and several diagnostic methods were used.

R: We are grateful with the Reviewer for the positive opinion of our work.

  1. In my opinion, additional laboratory markers should be considered as predictors.

R: We agree with the Reviewer that laboratory results may have a predictive value in the mortality risk assessment of STEMI patients. Accordingly, GFR and peak troponins were considered in the modified Table 3.

  1. Is it justified to consider glomerular filtration rate as a predictor of ventricular fibrillation?

R: We agree with the Reviewer that GFR on admission cannot be assessed as an early predictor of VF. Accordingly, we eliminated GFR from the modified Table 2 and we decided to consider only the variables available at admission (anamnestic variables, ECG features and Killip class).

  1. P-values should be consistently reported to 3 decimals.

R: We modified the manuscript according to the Reviewer’ suggestions.

  1. More than 70% of references are over 10 years old.

R: According to the Reviewer’ suggestions, we eliminated three old references and added one published in 2020 (ref #16). 

Reviewer 2 Report

In this interesting analysis, authors address a question whether certain specific ECG patterns in STEMI patients that may occur BEFORE stenting are: a) associated with the probability of primary ventricular fibrillation (PVF). This is of intereset, since PVF has been shown as an ominus sign - it is associated (predicts) higher 30-day mortality; b) associated with 30-day mortality.

In an attempt to answer these questions, the authors apply the following approach: a) first, they try to see whether any of the three specific ECG findings (Triangular-lambda; Tombstoning and giant R) is independently associated with occurrence of PVF; b) next, they try to see whether any of these ECG findings is independently associated with 30-day mortality.

As a result of the first-step analysis, they conclude that triangular-lambda pattern is associated with a higher risk of PVF.

As a result of the second-step analysis, they conclude that triangula-lambda, although associated with higher probability of PVF (which, in turn is known to be associated with higher 30-day mortality) is NOT associated (independently) with 30-day mortality.

There seems to be a certain conceptual/logical flaw in this process.

Namely, the implied sequence of events is:  a specific ECG pattern → results in higher risk of PVF → PVF results in higher 30-day mortality. In such a sequence, a specific ECG pattern is a "predictor". In the first step, it reflects on the probability of PVF, and then PVF reflects on the probability of death. Hence, PVF if is a "mediator" - ECG pattern reflects on death "through" its effect on PVF. My point is: the multivariate model shown in Table 4, where cardiac death is the outcome, SHOULD NOT CONTAIN ECG patterns as preditors. It should contain PVF. If triangular-lambda is a predictor of PVF, and PVF is a mediator, then "skipping" the mediator from the causal process could "delete" the association between the ECG pattern and death.

To try to sum-up. 1. The model in Table 3 tries to detect any of the 3 ECG patterns as predictive of PVF. If finds that TW-lambda is associated strongly with higher odds of PVF. Giant R and tomboning are not. 2. To evaluate whether, therefore, TW-lambda has a "link" to mortality, one would need to employ MEDIATION ANALYSIS. Mediation analysis (there are macros in SAS and in SPSS, and also programs in R to perform mediation analysis) first evaluates the link   predictor (TW-lambda) → PVF (mediator). In the next step, it evaluates the link  PVF (mediator)  → 30-day death (outcome). Finally, it calculates (i) total effect of predictor on the outcome and (ii) "mediated" or indirect effect of predictor on the outcome, i.e., it quantifies the extent of association between a predictor (TW-lambda) and the outcome (death) that "goes through" the effect of predictor on the mediator (PVF).

Namely, in the present form, the manuscript and the analysis say: Ok, PVF is relevant for 30-day survival. Let's see whether certain ECG pattern(s) are relevant for the risk of PVF. Then, an ECG pattern strongly associated with PVF is found. It is intuitive to think that - through PVF - it is then relevant for 30-day mortality. But the conclusion is - it is not. Why would it then be revelant at all? The fact is - that this lack of association between TW-lambda might actually be an artifact...and that there in fact IS an association between TW-lambda and mortality that goes through its effect on PVF - but for that, one needs to redefine the analysis.

Several further points to re-consider :

1. There were only 14 patients with TW-lambda pattern. This is a very small subset. allthough there were some 400 patients, a multivariate model may not ascertain control for confounding ...when there are many categorical variables in the model, one can get "cells" with actually no events, and then no reasonable adjustments are possible (e.g., there are no women (only a quaretr of all were women), with eGFR>60 with TW-lambda (only 14 subjects) and Killip class III (only 24 subjects). The authors should consider to use eGFR as continuous, and to dichotomize Killip class. 

2. In both models (Table 3, table 4), the 3 ECG patterns of interest should not be used as 3 distinct variables. There should be one variable termed "ECG pattern" with 4 levels: i. giant R, ii. TW-lambda, iii. tombstone, iv. none of those. Then each of the 3 of interest would be contrasted to the "reference" (none of those).

I believe that the authors should re-consider their data analysis before actual peer-review of the manuscript.

Author Response

  1. In this interesting analysis, authors address a question whether certain specific ECG patterns in STEMI patients that may occur BEFORE stenting are: a) associated with the probability of primary ventricular fibrillation (PVF). This is of interest, since PVF has been shown as an ominus sign - it is associated (predicts) higher 30-day mortality; b) associated with 30-day mortality. In an attempt to answer these questions, the authors apply the following approach: a) first, they try to see whether any of the three specific ECG findings (Triangular-lambda; Tombstoning and giant R) is independently associated with occurrence of PVF; b) next, they try to see whether any of these ECG findings is independently associated with 30- day mortality. As a result of the first-step analysis, they conclude that triangular-lambda pattern is associated with a higher risk of PVF. As a result of the second-step analysis, they conclude that triangular-lambda, although associated with higher probability of PVF (which, in turn is known to be associated with higher 30-day mortality) is NOT associated (independently) with 30-day mortality. There seems to be a certain conceptual/logical flaw in this process.

Namely, the implied sequence of events is: a specific ECG pattern → results in higher risk of PVF → PVF results in higher 30-day mortality. In such a sequence, a specific ECG pattern is a "predictor". In the first step, it reflects on the probability of PVF, and then PVF reflects on the probability of death. Hence, PVF if is a "mediator" – ECG pattern reflects on death "through" its effect on PVF. My point is: the multivariate model shown in Table 4, where cardiac death is the outcome, SHOULD NOT CONTAIN ECG patterns as predictors. It should contain PVF. If triangular-lambda is a predictor of PVF, and PVF is a mediator, then "skipping" the mediator from the causal process could "delete" the association between the ECG pattern and death.

To try to sum-up. 1. The model in Table 3 tries to detect any of the 3 ECG patterns as predictive of PVF. If finds that TW-lambda is associated strongly with higher odds of PVF. Giant R and tombstoning are not. 2. To evaluate whether, therefore, TW-lambda has a "link" to mortality, one would need to employ MEDIATION ANALYSIS. Mediation analysis (there are macros in SAS and in SPSS, and also programs in R to perform mediation analysis) first evaluates the link predictor (TW-lambda) → PVF (mediator). In the next step, it evaluates the link PVF (mediator) → 30-day death (outcome). Finally, it calculates (i) total effect of predictor on the outcome and (ii) "mediated" or indirect effect of predictor on the outcome, i.e., it quantifies the extent of association between a predictor (TW-lambda) and the outcome (death) that "goes through" the effect of predictor on the mediator (PVF). Namely, in the present form, the manuscript and the analysis say: Ok, PVF is relevant for 30-day survival. Let's see whether certain ECG pattern(s) are relevant for the risk of PVF. Then, an ECG pattern strongly associated with PVF is found. It is intuitive to think that - through PVF - it is then relevant for 30-day mortality. But the conclusion is – it is not. Why would it then be relevant at all? The fact is - that this lack of association between TW-lambda might actually be an artifact...and that there in fact IS an association between TW-lambda and mortality that goes through its effect on PVF - but for that, one needs to redefine the analysis.

            R: we thank the Reviewer for the clever suggestions. We extensively modified the statistical analysis in order to better present the study design and our findings.

  1. Several further points to re-consider:

a: There were only 14 patients with TW-lambda pattern. This is a very small subset. Although there were some 400 patients, a multivariate model may not ascertain control for confounding. The authors should consider to use eGFR as continuous, and to dichotomize Killip class.

R: we thank the Reviewer for the suggestion, which allows us to improve the manuscript quality. Thus, we modified the statistical analysis, as specified in the text, including, for each multivariate model, only the variables clinically meaningful in order to maintain a ≈1:10 co-variates to outcome ratio as to avoid the risk of overfitting.

b: In both models (Table 3, table 4), the 3 ECG patterns of interest should not be used as 3 distinct variables. There should be one variable termed "ECG pattern" with 4 levels: i. giant R, ii. TW-lambda, iii. tombstone, iv. none of those. Then each of the 3 of interest would be contrasted to the "reference" (none of those).

R:  we thank the Reviewer for the suggestion, which allows us to improve the manuscript quality.

Reviewer 3 Report

The authors describe the incidence of VFib and 30-days outcomes in STEMI patients. They find an association between ST-elevation pattern and incidnece of VFib but not 30 dyas prognosis. The study is well done the paper well written. 

What is the rationale of  combining TW and Lambda in a single group?

Which parameters were introduced in the univariate analyses? only those in the tables? why not culprit vessel, for instance?

Author Response

1. The authors describe the incidence of VFib and 30-days outcomes in STEMI patients. They find an association between ST-elevation pattern and incidnece of VFib but not 30 days prognosis. The study is well done the paper well written. 

R: We are grateful with the Reviewer for the positive opinion of our work.

2. What is the rationale of combining TW and Lambda in a single group?

R: Because TW and Lambda pattern have in common the steeply downsloping ST segment toward the T wave. In a paper by Aizawa et al (Ref #9), TW and Lambda pattern were both included in the Type I STEMI pattern.

3. Which parameters were introduced in the univariate analyses? only those in the tables? why not culprit vessel, for instance?

R: Only variables considered more relevant for the scope of the study were included into the models, as specified in the revised manuscript (methods and results parts).

Round 2

Reviewer 2 Report

In the revised version, after adopting the mediation analysis approach, authors were able to identify a link between a specific ECG pattern and 30-day mortality.

Further minor suggestions:

  1. There is a typo in units of eGFR (mq, instead of m2).
  2. In the reply to one of my comments, the authors elaborated the method of multivariate model building. This should be included in the Methods section. Brevity is important, but clarity is MORE important. Building multivariate models is a tricky business - and to supprot suggestions about some relationships in very complex settings (like this one)  - it is essential that every step/reasoning in model building is straightforwardly explained. It makes the case more convincing.
  3. A table or tables with multivariate models should be included in the manuscript.
  4. There is also a way how to report mediation analysis (there same general rules or principles) - so it would be only fair and decent to present mediation analysis.
  5. There are different R packages, there are SPSS and SAS macros for mediation. They do not all use the same approach etc. - hence, the authors should specifically declare the software/package/macro used.
  6. There was an indirect link through LVEF. I would not say that there was "no indirect link through PVF...- I would say that there was a clear tendency of some "mediation" - the point estimate is clearly above 0, and the  95% CIs are predominantly above 0, falling only in a minor part below 0 (p=0.108) - with some more events/subjects - it would have been above 0 entirely. Therefore, I would say that there was numerical tendency of some moderate indirect effect - but due to the event number/sample size limitations - there is quite some uncertainty about that.
  7. In reporting CIs for continuous variables, when lower limit is <0, a "-" sign between upper and lower limit is confusing (e.g., from -0.013 to 0.14 shown as -0.013-0.14...is confusing, use comma  e.g., -0.013, 0.14).

Author Response

Reviewer: In the revised version, after adopting the mediation analysis approach, authors were able to identify a link between a specific ECG pattern and 30-day mortality.

Further minor suggestions:

  1. There is a typo in units of eGFR (mq, instead of m2).

R: We thank the reviewer for the correction. Accordingly, we modified Table 4 replacing mq with m2.

2-3. In the reply to one of my comments, the authors elaborated the method of multivariate model building. This should be included in the Methods section. Brevity is important, but clarity is MORE important. Building multivariate models is a tricky business - and to supprot suggestions about some relationships in very complex settings (like this one)  - it is essential that every step/reasoning in model building is straightforwardly explained. It makes the case more convincing. A table or tables with multivariate models should be included in the manuscript.

R: We thank the reviewer for the comment, which allow us to clarify this important point of our work. In the revised version (Methods section, line 100-107), the regression building process has been reported. Furthermore, in the Results section (lines 172-174 and line 187), the selection steps used for choosing the variables included in the univariate and multivariate analysis have been specified. Finally, as suggested, Table 3 and Table 5 now report the results of Univariate and Multivariate analysis.

4-5-6. There is also a way how to report mediation analysis (there same general rules or principles) - so it would be only fair and decent to present mediation analysis. There are different R packages, there are SPSS and SAS macros for mediation. They do not all use the same approach etc. - hence, the authors should specifically declare the software/package/macro used. There was an indirect link through LVEF. I would not say that there was "no indirect link through PVF...- I would say that there was a clear tendency of some "mediation" - the point estimate is clearly above 0, and the  95% CIs are predominantly above 0, falling only in a minor part below 0 (p=0.108) - with some more events/subjects - it would have been above 0 entirely. Therefore, I would say that there was numerical tendency of some moderate indirect effect - but due to the event number/sample size limitations - there is quite some uncertainty about that.

R: We thank the Reviewer for the comment, and we kindly apologize for the inaccurate presentation of our results. Accordingly, in Method section, all the details about software and packages used for performing the analysis (line 109-110 and 112-115) have been added. Furthermore, results of mediation analysis have been more clearly reported (line 191-199, Table 6 and Table 7).

  1. In reporting CIs for continuous variables, when lower limit is <0, a "-" sign between upper and lower limit is confusing (e.g., from -0.013 to 0.14 shown as -0.013-0.14...is confusing, use comma  e.g., -0.013, 0.14).

R: We thank the reviewer for the suggestion. We modified the manuscript accordingly.